# MODELING LATENT ATTENTION WITHIN NEURAL NETWORKS

## ABSTRACT

Deep neural networks are able to solve tasks across a variety of domains and modalities of data. Despite many empirical successes, we lack the ability to clearly understand and interpret the learned mechanisms that contribute to such effective behaviors and more critically, failure modes. In this work, we present a general method for visualizing an arbitrary neural network's inner mechanisms and their power and limitations. Our dataset-centric method produces visualizations of how a trained network attends to components of its inputs. The computed "attention masks" support improved interpretability by highlighting which input attributes are critical in determining output. We demonstrate the effectiveness of our framework on a variety of deep neural network architectures in domains from computer vision and natural language processing. The primary contribution of our approach is an interpretable visualization of attention that provides unique insights into the network's underlying decision-making process irrespective of the data modality.

## 1 INTRODUCTION

Machine-learning systems are ubiquitous, even in safety-critical areas. Trained models used in self-driving cars, healthcare, and environmental science must not only strive to be error free but, in the face of failures, must be amenable to rapid diagnosis and recovery. This trend toward real-world applications is largely being driven by recent advances in the area of deep learning. Deep neural networks have achieved state-of-the-art performance on fundamental domains such as image classification (Krizhevsky et al., 2012), language modeling (Bengio et al., 2000; Mikolov et al., 2010), and reinforcement learning from raw pixels (Mnih et al., 2015). Unlike traditional linear models, deep neural networks offer the significant advantage of being able to learn their own feature representation for the completion of a given task. While learning such a representation removes the need for manual feature engineering and generally boosts performance, the resulting models are often hard to interpret, making it significantly more difficult to assign credit (or blame) to the model's behaviors. The use of deep learning models in increasingly important application areas underscores the need for techniques to gain insight into their failure modes, limitations, and decision-making mechanisms.

Substantial prior work investigates methods for increasing interpretability of these systems. One body of work focuses on visualizing various aspects of networks or their relationship to each datum they take as input Yosinski et al. (2015); Zeiler & Fergus (2015). Other work investigates algorithms for eliciting an explanation from trained machine-learning systems for each decision they make Ribeiro et al. (2016); Baehrens et al. (2010); Robnik-Šikonja & Kononenko (2008). A third line of work, of which our method is most aligned, seeks to capture and understand what networks focus on and what they ignore through *attention* mechanisms.

Attention-based approaches focus on network architectures that specifically attend to regions of their input space. These "explicit" attention mechanisms were developed primarily to improve network behavior, but additionally offer increased interpretability of network decision making through highlighting key attributes of the input data (Vinyals et al., 2015; Hermann et al., 2015; Oh et al., 2016; Kumar et al., 2016). Crucially, these explicit attention mechanisms act as filters on the input. As such, the filtered components of the input could be replaced with reasonably generated noise without dramatically affecting the final network output. The ability to selectively replace irrelevant components of the input space is a direct consequence of the explicit attention mechanism. The insight at the heart of the present work is that it is possible to evaluate the property of "selective

replaceability" to better understand a network that lacks any explicit attention mechanism. An architecture without explicit attention may still depend more on specific facets of its input data when constructing its learned, internal representation, resulting in a "latent" attention mechanism.

In this work, we propose a novel approach for indirectly measuring latent attention mechanisms in arbitrary neural networks using the notion of selective replaceability. Concretely, we learn an auxiliary, "Latent Attention Network" (LAN), that consumes an input data sample and generates a corresponding mask (of the same shape) indicating the degree to which each of the input's components are replaceable with noise. We train this LAN by corrupting the inputs to a pre-trained network according to generated LAN masks and observing the resulting corrupted outputs. We define a loss function that trades off maximizing the corruption of the input while minimizing the deviation between the outputs generated by the pre-trained network using the true and corrupted inputs, independently. The resultant LAN masks must learn to identify the components of the input data that are most critical to producing the existing network's output (*i.e.* those regions that are given the most attention by the existing network.)

We empirically demonstrate that the LAN framework can provide unique insights into the inner workings of various pre-trained networks. Specifically, we show that classifiers trained on a Translated MNIST domain learn a two-stage process of first localizing a digit within the image before determining its class. We use this interpretation to predict regions on the screen where digits are less likely to be properly classified. Additionally, we use our framework to visualize the latent attention mechanisms of classifiers on both image classification (to learn the visual features most important to the network's prediction), and natural language document classification domains (to identify the words most relevant to certain output classes). Finally, we examine techniques for generating attention masks for specific samples, illustrating the capability of our approach to highlight salient features in individual members of a dataset.

## 2 RELATED WORK

We now survey relevant literature focused on understanding deep neural networks, with a special focus on approaches that make use of attention.

Attention has primarily been applied to neural networks to improve performance Mnih et al. (2014); Gregor et al. (2015); Bahdanau et al. (2014). Typically, the added attention scheme provides an informative prior that can ease the burden of learning a complex, highly structured output space (as in machine translation). For instance, Cho et al. (2015) survey existing *content-based* attention models to improve performance in a variety of supervised learning tasks, including speech recognition, machine translation, image caption generation, and more. Similarly, Yang et al. (2016) apply stacked attention networks to better answer natural language questions about images, and Goyal et al. (2016) investigate a complementary method for networks specifically designed to answer questions about visual content; their approach visualizes which content in the image is used to inform the network's answer. They use a strategy similar to that of attention to visualize what a network focuses on when tasked with visual question answering problems.

Yosinski et al. (2015) highlight an important distinction for techniques that visualize aspects of networks: *dataset-centric* methods, which require a trained network and data for that network, and *network-centric* methods, which target visualizing aspects of the network independent of any data. In general, dataset-centric methods for visualization have the distinct advantage of being network agnostic. Namely, they can treat the network to visualize entirely as a black box. All prior work for visualizing networks, of both dataset-centric and network-centric methodologies, is specific to particular network architectures (such as convolutional networks). For example, Zeiler & Fergus (2015) introduce a visualization method for convolutional neural networks (CNNs) that illustrates which input patterns activate feature maps at each layer of the network. Their core methodology is to project activations of nodes at any layer of the network back to the input pixel space using a Deconvolutional Network introduced by Zeiler et al. (2011), resulting in highly interpretable feature visualizations. An exciting line of work has continued advancing these methods, as in Nguyen et al. (2016); Simonyan et al. (2013), building on the earlier work of Erhan et al. (2009) and Berkes & Wiskott (2005).

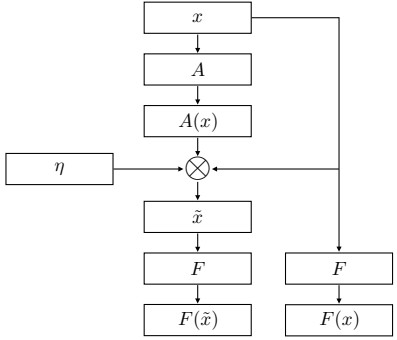

Figure 1: Diagram of the Latent Attention Network (LAN) framework.

A different line of work focuses on strategies for eliciting explanations from machine learning systems to increase interpretability Ribeiro et al. (2016); Baehrens et al. (2010); Robnik-Šikonja & Kononenko (2008). Lei et al. (2016) forces networks to output a short "rationale" that (ideally) justifies the network's decision in Natural Language Processing tasks. Bahdanau et al. (2014) advance a similar technique in which neural translation training is augmented by incentivizing networks to jointly align *and* translate source texts. Lastly, Zintgraf et al. (2017) describe a method for eliciting visualizations that offer explanation for decisions made by networks by highlighting regions of the input that are considered evidence for or against a particular decision.

In contrast to all of the discussed methods, we develop a dataset-centric method for visualizing attention in an *arbitrary* network architecture. To the best of our knowledge, the approach we develop is the first of its kind in this regard. One similar class of methods is sensitivity analysis, introduced by Garson (1991), which seeks to understand input variables' contribution to decisions made by the network Wang et al. (2000); Gedeon (1997); Gevrey et al. (2006). Sensitivity analysis has known limitations Mazurowski & Szecowka (2006), including failures in highly dependent input spaces and restriction to ordered, quantitative input spaces Montano & Palmer (2003).

## 3 METHOD

A key distinguishing feature of our approach is that we assume minimal knowledge about the network to be visualized. We only require that the network $F : \mathbb{R}^d \mapsto \mathbb{R}^\ell$ be provided as a black-box function (that is, we can provide input $x$ to $F$ and obtain output $F(x)$) through which gradients can be computed. Since we do not have access to the network architecture, we can only probe the network either at its input or its output. In particular, our strategy is to modify the input by selectively replacing components via an attention mask, produced by a learned Latent Attention Network (LAN).

### 3.1 LATENT ATTENTION NETWORK FRAMEWORK

A Latent Attention Network is a function $A : \mathbb{R}^d \mapsto [0, 1]^d$ that, given an input $x$ (for the original network $F$), produces an *attention mask* $A(x)$ of the same shape as $x$. The attention mask seeks to identify input components of $x$ that are critical to producing the output $F(x)$. Equivalently, the attention mask determines the degree to which each component of $x$ can be corrupted by noise while minimally affecting $F(x)$. To formalize this notion, we need two additional *design components*:

$$\mathcal{L}_F : \mathbb{R}^\ell \times \mathbb{R}^\ell \mapsto \mathbb{R} \quad \text{a loss function in the output space of } F,$$
$$H : \mathbb{R}^d \mapsto \mathbb{R} \quad \text{a noise probability density over the input space of } F. \tag{1}$$

We can now complete the specification of the LAN framework. As illustrated in Figure 1, given an input $x$, we draw a noisy vector $\eta \sim H$ and corrupt $x$ according to $A(x)$ as follows:

$$\tilde{x} = A(x) \cdot \eta + (\mathbf{1} - A(x)) \cdot x, \tag{2}$$

where $\mathbf{1}$ denotes a tensor of ones with the same shape as $A(x)$, and all operations are performed element-wise. Under this definition of $\tilde{x}$, the components of $A(x)$ that are close to 0 indicate that

the corresponding components of $x$ represent signal/importance, and those close to 1 represent noise/irrelevance. Finally, we can apply the black-box network $F$ to $\tilde{x}$ and compare the output $F(\tilde{x})$ to the original $F(x)$ using the loss function $\mathcal{L}_F$.

An ideal attention mask $A(x)$ replaces/corrupts as many input components as possible (it has $A(x)$ components close to 1), while minimally distorting the original output $F(x)$, as measured by $\mathcal{L}_F$. Hence we train the LAN $A$ by minimizing the following training objective for each input $x$:

$$\mathcal{L}_{\text{LAN}}(x) = \mathbb{E}_{\eta \sim H} \left[ \mathcal{L}_F(F(\tilde{x}), F(x)) - \beta \overline{A(x)} \right], \tag{3}$$

where $\overline{A(x)}$ denotes the mean value of the attention mask for a given input, $\tilde{x}$ is a function of both $\eta$ and $A(x)$ as in Equation 2, and $\beta > 0$ is a hyperparameter for weighting the amount of corruption applied to the input against the reproducibility error with respect to $\mathcal{L}_F$, for more information about this trade-off see Section E in the Appendix.

## 3.2 LATENT ATTENTION NETWORK DESIGN

To specify a LAN, we provide two components: the loss function $\mathcal{L}_F$ and the noise distribution $H$. The choice of these two components depends on the particular visualization task. Typically, the loss function $\mathcal{L}_F$ is the same as the one used to train $F$ itself, although it is not necessary. For example, if a network $F$ was pre-trained on some original task but later applied as a black-box within some novel task, one may wish to visualize the latent attention with respect to the new task's loss to verify that $F$ is considering expected parts of the input.

The noise distribution $H$ should reflect the expected space of inputs to $F$, since input components' importance is measured with respect to variation determined by $H$. In the general setting, $H$ could be a uniform distribution over $\mathbb{R}^d$; however, we often operate in significantly more structured spaces (e.g. images, text). In these structured cases, we suspect it is important to ensure that the noise vector $\eta$ lies near the manifold of the input samples.

Based on this principle, we propose two methods of defining $H$ via the generating process for $\eta$:

- Constant noise $\eta_{\text{const}}$: In domains where input features represent quantities with default value $c$ (e.g. 0 word counts in a bag of words, 0 binary valued images), set $\eta = c\mathbf{1}$, where $\mathbf{1}$ is a tensor of ones with the appropriate shape and $c \in \mathbb{R}$.
- Bootstrapped noise $\eta_{\text{boot}}$: Draw uniform random samples from the training dataset.

We expect that the latter approach is particularly effective in domains where the data occupies a small manifold of the input space. For example, consider that that the set of natural images is much smaller than the set of possible images. Randomly selecting an image guarantees that we will be near that manifold, whereas other basic forms of randomness are unlikely to have this property.

## 3.3 SAMPLE-SPECIFIC LATENT ATTENTION MASKS

In addition to optimizing whole networks that map arbitrary inputs to attention masks, we can also directly estimate that attention-scheme of a single input. This sample-specific approach simplifies a LAN from a whole network to just a single, trainable variable that is the same shape as the input. This translates to the following optimization procedure:

$$\mathcal{L}_{\text{SSL}}^x = \mathbb{E}_{\eta \sim H} \left[ \mathcal{L}_F(F(\tilde{x}), F(x)) - \beta \overline{A(x)} \right] \tag{4}$$

where $\overline{A(x)}$ represents the attention mask learned specifically for sample $x$ and $\tilde{x}$ is a function of $\eta$, $A$ and $x$ defined in Eq. (2).

## 4 EXPERIMENTS

To illustrate the wide applicability of the LAN framework, we conduct experiments in a variety of typical learning tasks, including digit classification and object classification in natural images. The goal of these experiments is to demonstrate the effectiveness of LANs to visualize latent

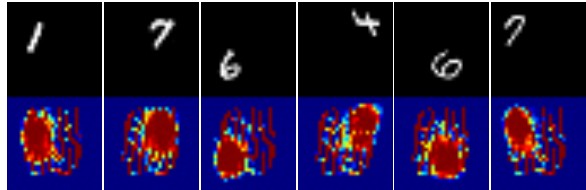

Figure 2: Visualization of attention maps for different translated MNIST digits. For each pair of images, the original translated MNIST digit is displayed on the top, and a visualization of the attention map is displayed on the bottom (where warmer colors indicate more important regions to the pre-trained classifier). Notice the blobs of network importance around each digit, and the seemingly constant "griding" pattern present in each of the samples.

attention mechanisms of different network types. Additionally, we conduct an experiment in a topic-modeling task to demonstrate the flexibility of LANs across multiple modalities. While LANs can be implemented with arbitrary network architectures, we restrict our focus here to fully-connected LANs and leave investigations of more expressive LAN architectures to future work. More specifically, our LAN implementations range from 2–5 fully-connected layers each with fewer than 1000 hidden units. At a high level, these tasks are as follows (see supplementary material for training details):

**Translated MNIST**

*Data* : A dataset of $28 \times 28$ grayscale images with MNIST digits, scaled down to $12 \times 12$, are placed in random locations. No modifications are made to the orientation of the digits.

*Task* : We train a standard deep network for digit classification.

**CIFAR-10**

*Data* : A dataset of 3-channel $32 \times 32$ color images of objects or animals, each belonging to one of ten unique classes. The images are typically centered around the classes they depict.

*Task* : We train a standard CNN for object detection.

**Newsgroup-20**

*Data* : A dataset consisting of news articles belonging to one of twenty different topics. The list of topics includes politics, electronics, space, and religion, amongst others.

*Task* : We train a bag-of-words neural network, similar to the Deep Averaging Network (DAN) of Iyyer et al. (2015) to classify documents into one of the twenty different categories.

For each experiment, we train a network $F$ (designed for the given task) to convergence. Then, we train a Latent Attention Network, $A$ on $F$. For all experiments conducted with image data, we used bootstrapped noise while our exploratory experiment with natural language used constant noise. Since LANs capture attention in the input space, the result of the latter training procedure is to visualize the attention mechanism of $F$ on any sample in the input. For a detailed description of all experiments and associated network architectures, please consult the supplementary material.

## 5 RESULTS

### 5.1 TRANSLATED MNIST RESULTS

Results are shown in Figure 2. We provide side-by-side visualizations of samples from the Translated MNIST dataset and their corresponding attention maps produced by the LAN network. In these attention maps, there are two striking features: (1) a blob of attention surrounding the digit and (2) an unchanging grid pattern across the background. This grid pattern is depicted in Figure 3a.

In what follows, we support an interpretation of the grid effect illustrated in Figure 3a. Through subsequent experiments, we demonstrate that our attention masks have illustrated that the classifier network operates in two distinct phases:

1. Detect the presence of a digit somewhere in the input space.

2. Direct attention to the region in which the digit was found to determine its class.

Under this interpretation, one would expect classification accuracy to decrease in regions not spanned by the constant grid pattern. To test this idea, we estimated the error of the classifier on digits centered at various locations in the image. We rescaled the digits to $7 \times 7$ pixels to make it easier to fit them in the regions not spanned by the constant grid. Visualizations of the resulting accuracies are displayed in Figure 3b. Notice how the normalized accuracy falls off around the edges of the image (where the constant grid is least present). This effect is particularly pronounced with smaller digits, which would be harder to detect with a fixed detection grid.

To further corroborate our hypothesis, we conducted an additional experiment with a modified version of the Translated MNIST domain. In this new domain, digits are scaled to $12 \times 12$ pixels and never occur in the bottom right $12 \times 12$ region of the image. Under these conditions, we retrained our classifier and LAN, obtaining the visualization of the constant grid pattern and probability representation presented in Figure 3(c-d). Notice how the grid pattern is absent from the bottom right-hand corner where digits never appeared at training time. Consequently, the accuracy of the classifier falls off if tested on digits in this region.

Through these results, we showcase the capability of LANs to produce attention masks that not only provide insights into the inner workings of a trained network but also serve as a diagnostic for predicting likely failure modes.

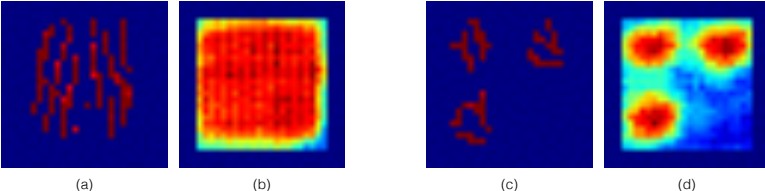

Figure 3: (a) Constant grid pattern observed in the attention masks on Translated MNIST. (b) Accuracy of the pre-trained classifier on $7 \times 7$ digits centered at different pixels. Each pixel in the images is colored according to the estimated normalized accuracy on digits centered at that pixel where warmer colors indicate higher normalized accuracy. Only pixels that correspond to a possible digit center are represented in these images, with other pixels colored dark blue. (c–d) Duplicate of (a) and (b) for a pre-trained network on a modified Translated MNIST domain where no digits can appear in the bottom right hand corner.

## 5.2 CIFAR-10 CNN

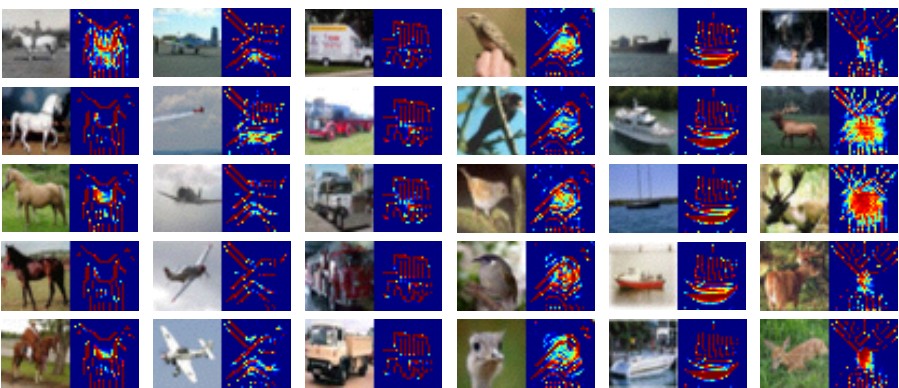

Figure 4: Each frame pairs an input image (left) with its LAN attention mask (right). Each column represents a different category: horse, plane, truck, bird, ship, and deer.

In Figure 4, we provide samples of original images from the CIFAR-10 dataset alongside the corresponding attention masks produced by the LAN. Notice that, for images belonging to the same class, the resulting masks capture common visual features such as tail feathers for birds or hulls/masts for ships. The presence of these features in the mask suggests that the underlying classifier learns a canonical representation of each class to discriminate between images and to confirm its classification. We further note that, in addition to revealing high level concepts in the learned classifier, the LAN appears to demonstrate the ability to compose those concepts so as to discriminate between classes. This property is most apparent between the horse and deer classes, both of which show extremely similar regions of attention for capturing legs while deviating in their structure to confirm the presence of a heads or antlers, respectively.

## 5.3 Newsgroup-20 Document Classification Results

Tables 1 and 2 contrast words present in documents against the 15 most important words, as determined by the corresponding attention mask, for topic classification. We note that these important words generally tend to be either in the document itself (highlighted in yellow) or closely associated with the category that the document belongs to. The absence of important words from other classes is explained by our choice of $\eta_0$-noise, which produces more visually appealing attention-masks, but doesn't penalize the LAN for ignoring such words. We suspect that category-associated words not present in the document occur due to the capacity limitations on the fully-connected LAN architecture on a high dimensional and poorly structured bag-of-words input space. Future work will further explore the use of LANs in natural language tasks.

| Document Topic | Document Words (Unordered) | 15 Most Important Words |
|---|---|---|
| comp.sys.mac.hardware | ralph, rutgers, rom, univ, mac, gonzalez, gandalf, work, use, you, phone, drives, internet, camden, party, floppy, science, edu, roms, drive, upgrade, disks, computer | mac, drive, computer, problem, can, this, drives, disk, use, controller, UNK memory, for, boot, fax |

Table 1: A visualization of the attention mask generated for a specific document in the Newsgroup-20 Dataset. The document consists of the words above, and is labeled under the category "comp.sys.mac.hardware" which consists about topics relating to Apple Macintosh computer hardware. Note the top 15 words identified by the LAN Mask, and how they seem to be picking important words relevant to the true class of the given document.

| Document Topic | Document Words (Unordered) | 15 Most Important Words |
|---|---|---|
| soc.religion.christian | UNK, death, university, point, complaining, atheists, acs, isn, since, doesn, never, that, matters, god, incestuous, atterlep, rejection, forever, hell, step, based, talk, vela, eternal, edu, asked, worse, you, tread, will, not, and, rochester, fear, opinions, die, faith, fact, earth oakland, lot, don, christians, alan, melissa, rushing, angels, comparison, heaven, terlep | UNK, clh, jesus, this church, christians, interested, lord, christian, answer, will, heaven, find, worship, light |

Table 2: Another visualization of the attention mask generated for a specific document in the Newsgroup-20 Dataset. This document consists of the words above, and is labeled under the category "soc.religion.christian", which consists of topics relating to Christianity. The presence of UNK as an important word in this religious documents could be attributed to a statistically significant number of references to people and places from Abrahamic texts which are converted to UNK due to their relative uncommonness in the other document classes.

## 5.4 Sample-Specific Attention Masks

In all of the previous results, there is a strong sense in which the resultant attention masks are highly correlated with the pre-trained network outputs and less sensitive to variations in the individual input samples. Here we present results on the same datasets (see Figures 5, 3 and 4) using the

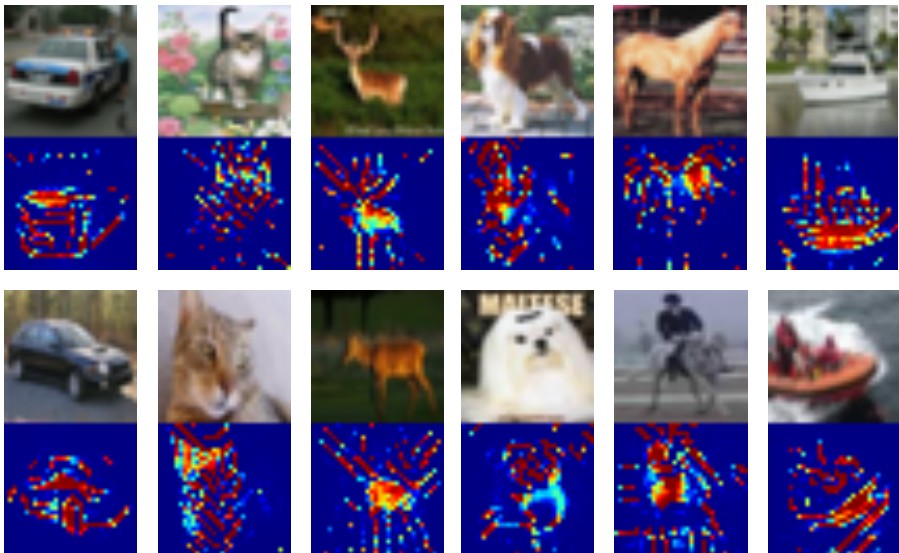

Figure 5: Each image pair contains a CIFAR-10 image and its corresponding sample-specific attention mask. Each column contains images from a different category: car, cat deer, dog, horse and ship. Notice how these sample-specific attention masks retain the class specific features mentioned in Section 5.2 while more closely tracking the subjects of the images.

sample specific objective defined in Eq. (4). We notice that these learned attention masks are more representative of nuances present in each invididual sample. This increase in contained information seems reasonable when considering the comparative ease of optimizing a single attention mask for a single sample rather than a full LAN that must learn to map from all inputs to their corresponding attention masks.

| Document Topic | Document Words (Unordered) | 15 Most Important Words |
|---|---|---|
| comp.sys.ibm.pc.hardware | UNK, video, chip, used, color, card, washington, drivers, name, edu, driver, chipset, suffice, functions, for, type, cica | card, chip, video, drivers, driver, type, used, cica, edu, washington, bike, functions, time, sale, color |

Table 3: A visualization of the sample specific attention mask generated for a specific document in the Newsgroup-20 Dataset. The document consists of the words above and is labeled under the category "comp.sys.ibm.pc.hardware" which consists of topics relating to personal computing and hardware. Words that are both in the document and detected by the sample specific attention mask are highlighted in yellow.

| Document Topic | Document Words (Unordered) | 15 Most Important Words |
|---|---|---|
| talk.religion.misc | newton, jesus, spread, died, writes, truth, ignorance, bliss, sandvik, not, strength, article, that, good, apple, kent | ignorance, died, sandvik, kent, newton, bliss, jesus, truth, good, can, strength, for, writes, computer, article |

Table 4: A visualization of the sample specific attention mask generated for a specific document in the Newsgroup-20 Dataset. The document consists of the words above and is labeled under the category "talk.religion.misc" which consists of topics relating to religion. Words that are both in the document and detected by the sample specific attention mask are highlighted in yellow.

## 6 CONCLUSION

As deep neural networks continue to find application to a growing collection of tasks, understanding their decision-making processes becomes increasingly important. Furthermore, as this space of tasks grows to include areas where there is a small margin for error, the ability to explore and diagnose problems within erroneous models becomes crucial.

In this work, we proposed Latent Attention Networks as a framework for capturing the latent attention mechanisms of arbitrary neural networks that draws parallels between noise-based input corruption and attention. We have shown that the analysis of these attention measurements can effectively diagnose failure modes in pre-trained networks and provide unique perspectives on the mechanism by which arbitrary networks perform their designated tasks.

We believe there are several interesting research directions that arise from our framework. First, there are interesting parallels between this work and the popular Generative Adversarial Networks (Goodfellow et al., 2014). It may be possible to simultaneously train $F$ and $A$ as adversaries. Since both $F$ and $A$ are differentiable, one could potentially exploit this property and use $A$ to encourage a specific attention mechanism on $F$, speeding up learning in challenging domains and otherwise allowing for novel interactions between deep networks. Furthermore, we explored two types of noise for input corruption: $\eta_{\text{const}}$ and $\eta_{\text{boot}}$. It may be possible to make the process of generating noise a part of the network itself by learning a nonlinear transformation and applying it to some standard variety of noise (such as Normal or Uniform). Since our method depends on being able to sample noise that is similar to the "background noise" of the domain, better mechanisms for capturing noise could potentially enhance the LAN's ability to pick out regions of attention and eliminate the need for choosing a specific type of noise at design time. Doing so would allow the LAN to pick up more specific features of the input space that are relevant to the decision-making process of arbitrary classifier networks.

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

We now describe each experiment in greater detail.

## A    TRANSLATED MNIST HANDWRITTEN DIGIT CLASSIFIER

Here we investigate the attention masks produced by a LAN trained on a digit classifier. We show how LANs provide intuition about the particular method a neural network uses to complete its task and highlight failure-modes. Specifically, we construct a "translated MNIST" domain, where the original digits are scaled down from $28 \times 28$ to $12 \times 12$ and positioned in random locations in the original $28 \times 28$ image. The network $F$ is a classifier, outputting the probability of each digit being present in a given image.

The pre-trained network, $F$ has the following architecture: $\text{Conv}(10, 2, (4 \times 4), \ell\text{-ReLU})$, $\text{Conv}(20, 2, (4 \times 4), \ell\text{-ReLU})$, $\text{FC}(10, \text{softmax})$. $F$ is trained with the Adam Optimizer for $100,000$ iterations with a learning rate of $0.001$ and with $\mathcal{L}_F = -\sum_i y_i \log F(x)_i$ where $y \in \mathbb{R}^{10}$ is a one-hot vector indicating the digit class.

The latent attention network, $A$ has the following architecture: $\text{FC}(100, \ell\text{-ReLU})$, $\text{FC}(784, \text{sigmoid})$, with its output being reshaped to a $28 \times 28$ image. $A$ is trained with the Adam Optimizer for $100,000$ iterations with a learning rate of $0.0001$. We use $\beta = 5.0$ and $\eta = \eta_{\text{boot}}$ for this experiment.

## B    CIFAR-10 CNN

In this experiment we demonstrate that the LAN framework can illuminate the decision making of classifier (based on the Alexnet architecture) on natural images. To avoid overfitting, we augment the CIFAR-10 dataset by applying small random affine transformations to the images at train time. We used $\beta = 5.0$ for this experiment.

The pre-trained network, $F$ has the following architecture: $\text{Conv}(64, 2, (5 \times 5), \ell\text{-ReLU})$, $\text{Conv}(64, 2, (5 \times 5), \ell\text{-ReLU})$, $\text{Conv}(64, 1, (3 \times 3), \ell\text{-ReLU})$, $\text{Conv}(64, 1, (3 \times 3), \ell\text{-ReLU})$, $\text{Conv}(32, 2, (3 \times 3), \ell\text{-ReLU})$, $\text{FC}(384, \text{tanh})$, $\text{FC}(192, \text{tanh})$, $\text{FC}(10, \text{softmax})$, where dropout and local response normalization is applied at each layer. $F$ is trained with the Adam Optimizer for $250,000$ iterations with a learning rate of $0.0001$ and with $\mathcal{L}_F = -\sum_i y_i \log F(x)_i$ where $y \in \mathbb{R}^{20}$ is a one-hot vector indicating the image class.

The latent attention network, $A$ has the following architecture: $\text{FC}(500, \ell\text{-ReLU})$, $\text{FC}(500, \ell\text{-ReLU})$, $\text{FC}(500, \ell\text{-ReLU})$, $\text{FC}(1024, \text{sigmoid})$, with its output being reshaped to a $32 \times 32 \times 1$ image and tiled 3 times on the channel dimension to produce a mask over the pixels. $A$ is trained with the Adam Optimizer for $250,000$ iterations with a learning rate of $0.0005$. We used $\beta = 7.0$ and $\eta = \eta_{\text{boot}}$ for this experiment.

## C    20 NEWSGROUPS DOCUMENT CLASSIFICATION

In this experiment, we extend the LAN framework for use on non-visual tasks. Namely, we show that it can be used to provide insight into the decision-making process of a bag-of-words document classifier, and identify individual words in a document that inform its predicted class label.

To do this, we train a Deep Averaging Network (DAN) (Iyyer et al., 2015) for classifying documents from the Newsgroup-20 dataset. The 20 Newsgroups Dataset consists of 18,821 total documents, partioned into a training set of 11,293 documents, and a test set of 7,528 documents. Each document belongs to 1 of 20 different categories, including topics in religion, sports, computer hardware, and

politics, to name a few. In our experiments, we utilize the version of the dataset with stop words (common words like "the", "his", "her") removed.

The DAN Architecture is very simple - each document is represented with a bag-of-words histogram vector, with dimension equal to the number of unique words in the dataset (the size of the vocabulary). This bag of words vector is then multiplied with an embedding matrix and divided by the number of words in the document, to generate a low-dimension normalized representation. This vector is then passed through two separate hidden layers (with dropout), and then a final softmax layer, to produce a distribution over the 20 possible classes. In our experiments we use an embedding size of 50, and hidden layer sizes of 200 and 150 units, respectively. We train the model for 1,000,000 mini-batches, with a batch size of 32. Like with our previous experiments, we utilize the Adam Optimizer Kingma & Ba (2014), with a learning rate of 0.00005.

The latent attention network, $A$ has the following architecture: $FC(100, \ell\text{-ReLU})$, $FC(1000, \ell\text{-ReLU})$, $FC(\text{vocab-size}, \text{sigmoid})$. $A$ is trained with the Adam Optimizer for $100,000$ iterations with a learning rate of $0.001$. We used $\beta = 50.0$ and $\eta = \eta_{\text{const}}$ with a constant value of 0.

## D    SAMPLE SPECIFIC EXPERIMENTS

In the sample specific experiments the same pre-trained networks are used as in the standard CIFAR-10 and Newsgroup-20 experiments. To train the sample specific masks, we used a learning rate of 0.001 and 0.05 for the Newsgroup-20 and CIFAR-10 experiments respectively. Both experiments used the Adam Optimizer and each mask is trained for $10,000$ iterations. We used $\beta = 50.0$ and $\eta = \eta_{\text{boot}}$ for both experiments.

## E    BETA HYPERPARAMETERS

In this section, we illustrate the role of the $\beta$ hyperparameter in the sample specific experiments. As stated earlier, $\beta$ controls the trade-off between the amount of corruption in the input and the similarity of the corrupted and uncorrupted inputs (measured as a function of the respective pre-trained network outputs). Based on the loss functions presented in equations 3 and 4, high values of $\beta$ encourage a larger amount of input corruption and weight the corresponding term more heavily in the loss computation. Intuitively, we would like to identify the minimal number of dimensions in the input space that most critically affect the output of the pre-trained network. The small amount of input corruption that corresponds to a low value of $\beta$ would make the problem of reconstructing pre-trained network outputs too simple, resulting in attention masks that deem all or nearly all input dimensions as important; conversely, a heavily corrupted input from a high value of $\beta$ could make output reconstruction impossible. We illustrate this for individual images from the CIFAR-10 dataset below:

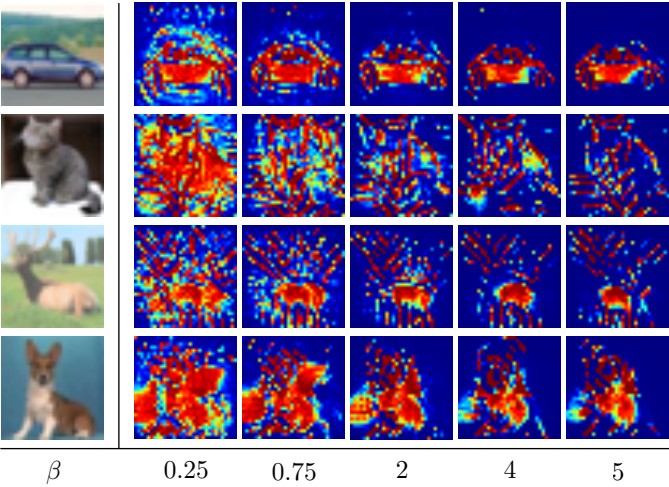

## F    IMAGENET RESULTS

To demonstrate the capacity of our technique to scale up, we visualize attention masks learned on top of the Inception network architecture introduced in Szegedy et al. (2015) and trained for the ILSVRC 2014 image classification challenge. We utilize a publicly available Tensorflow implementation of the pre-trained model[1]. In these experiments we learned sample-specific attention masks for different settings of $\beta$ on the images shown below. For our input corruption, we used uniformly sampled RGB noise: $\eta \sim \text{Uniform}(0, 1)$. We note that the attention masks produced on this domain seem to produce much richer patterns of contours than in the CIFAR-10 experiments. We attribute this difference to both the increased size of the images and the increased number of classes between the two problems(1000 vs 10).

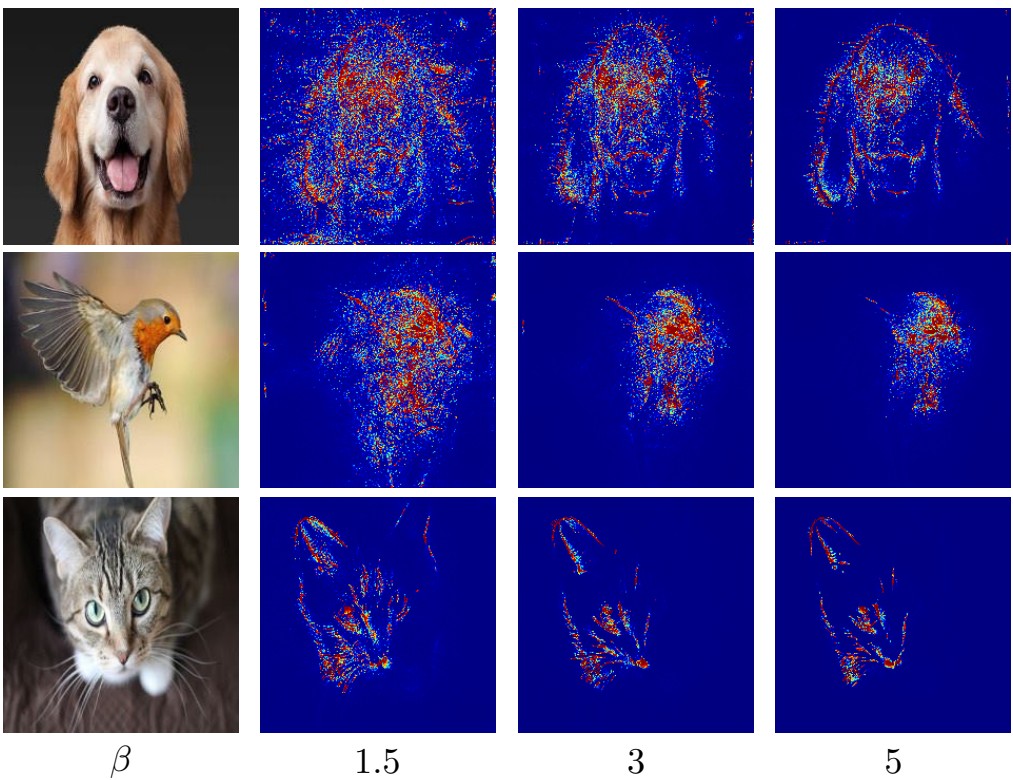

| $\beta$ | 1.5 | 3 | 5 |

## G    MOTIVATING EXAMPLE: TANK DETECTION

In this section, we motivate our approach by demonstrating how it identifies failure modes within a real-world application of machine learning techniques.

There is a popular and (allegedly) apocryphal story, relayed in Yudkowsky (2008), that revolves around the efforts of a US military branch to use neural networks for the detection of camouflaged tanks within forests. Naturally, the researchers tasked with this binary classification problem collected images of camouflaged tanks and empty forests in order to compile a dataset. After training, the neural network model performed well on the testing data and yet, when independently evaluated by other government agencies, failed to achieve performance better than random chance. It was later determined that the original dataset only collected positive tank examples on cloudy days leading to a classifier that discriminated based on weather patterns rather than the presence of camouflaged tanks.

We design a simple domain for highlighting the effectiveness of our approach, using the aforementioned story as motivation. The problem objective is to train an image classifier for detecting the presence of tanks in forests. Our dataset is composed of synthetic images generated through the

---

[1]https://github.com/tensorflow/models/tree/master/research/slim

random placement of trees, clouds and tanks. A representative image from this dataset is provided below on the left with its component objects highlighted and displayed on the right:

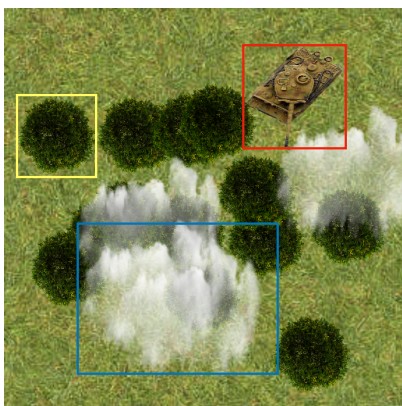
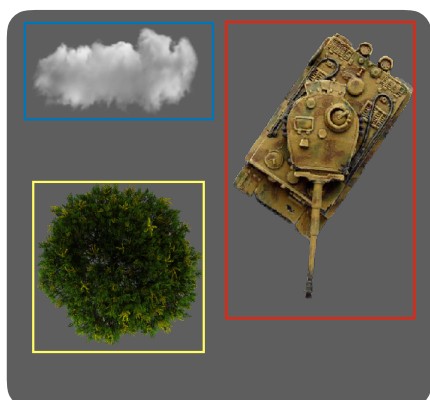

As in the story, our training and testing datasets are generated such that clouds are only present in positive examples of camouflaged tanks. A simple convolutional neural network architecture is then trained on this data and treated as the pre-trained network in our LAN framework. Unsurprisingly, this classifier suffers from the same problems outlined in Yudkowsky (2008); despite high accuracy on testing data, the classifier fails to detect tanks without the presence of clouds.

We now observe the sample-specific attention masks trained from this classifier:

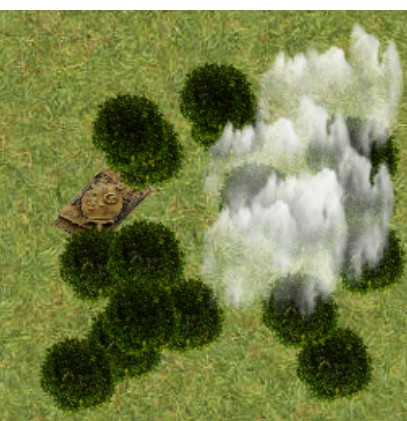
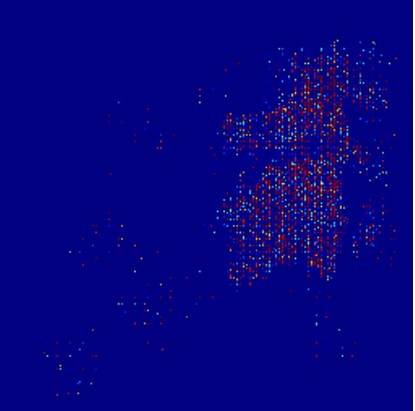

The resulting attention mask ($\beta = 1.0$ with bootstrapped noise $\eta = \eta_{\text{boot}}$) assigns high importance to the pixels associated with the cloud while giving no importance to the region of the image containing the tank. With this example, we underscore the utility of our methods in providing a means of visualizing the underlying "rationale" of a network for producing a given output. Our attention mask help recognize that the classifier's basis for detecting tanks is incorrectly based on the presence of clouds.

