# OpenReview forum: "Modeling Latent Attention Within Neural Networks"
_ICLR.cc/2018/Conference — Reject_

### Official Review · AnonReviewer3 · 2017-11-28
**This paper presented a general method for visualizing an arbitrary neural network's inner mechanisms.**

**Rating:** 4
**Confidence:** 4

**Review:**

The main contribution of the paper is to propose to learn a Latent Attention Network (LAN) that can help to visualize the inner structure of a deep neural network. To this end, the paper propose a novel training objective that can learn to tell the importance of each dimension of input. It is very interesting. However, one question is what is the potential usage of the model? Since the model need to train an another network to visualize the structure of a trained neural network, it is expensive, and I don't think the model can help use to design a better structure (at least the experiments did not show this point). And maybe different structures of LAN will produce different understanding of the trained model. Hence people are still not sure what kind of structure is the most helpful.

---

> ### Author Response · Authors · 2017-12-27
> **Usage and Cost of Modeling Latent Attention**
>
> Before directly addressing their concerns, we direct the reviewer’s attention to the newly added Sections E, F, & G in the supplementary material of our revised paper draft. These sections include new experiments that illustrate the effect of varying the beta hyperparameter, demonstrate the strength of our approach on the larger scale Inception network for the ILSVRC 2014 classification challenge, and further highlight the effectiveness of our approach in diagnosing model failure modes.
>
> The purpose of a LAN is to diagnose failure modes in trained neural networks. Consider a neural network that must make inferences in the healthcare domain, potentially having direct and immediate consequence to patients. When such a network makes any inference, it is paramount that there is an understanding of *why*. Identifying life-threatening illnesses or selecting an optimal course of treatment are just a few examples of decisions that must be made with as much transparency as possible. In capturing the importance in each dimension of a network’s input, our attention masks are a step forward in this direction of interpretable and understandable models. Please consult Section G of our supplementary materials for a demonstration of how sample-specific attention masks successfully identify model failure modes.
>
> With regards to the cost of a LAN, we note that the training time is itself a sunk cost that offers the potential of yielding information about a potentially erroneous model. In our work, we outline  two methods for learning attention masks. While one prescribes an entirely new network to train a mapping from input to attention masks, the other simply learns the attention mask for a single input directly. While we make no claims on the ability for a LAN to deliver a novel architecture that could remedy the problems of an existing network, it can certainly identify failure points from the dataset and provide sufficient motivation for abandoning an existing network architecture.
>
> Since the training objective would remain unchanged, we currently do not believe that using different structures for the LAN will result in significantly different attention masks. That said, the prohibitive cost of searching over the space of possible LAN architectures is, in part, mitigated by our second “sample-specific” approach for learning attention masks directly without constructing an entirely new network.

---

### Official Review · AnonReviewer1 · 2017-11-28

**Rating:** 5
**Confidence:** 4

**Review:**

The authors of this paper proposed a data-driven black-box visualization scheme. The paper primarily focuses on neural network models in the experiment section. The proposed method iteratively optimize learnable masks for each training example to find the most relevant content in the input that was "attended" by the neural network.  The authors empirically demonstrated their method on image and text classification tasks.

Strength:
           - The paper is well-written and easy to follow.
           - The qualitative analysis of the experimental results nicely illustrated how the learnt latent attention masks match with our intuition about how neural networks make its classification predictions.

        Weakness:
           - Most of the experiments in the paper are performed on small neural networks and simple datesets. I found the method will be more compiling if the authors can show visualization results on ImageNet models. Besides simple object recognition tasks, other more interesting tasks to test out the proposed visualization method are object detection models like end-to-end fast R-CNN, video classification models, and image-captioning models. Overall, the current set of experiments are limited to showcase the effectiveness of the proposed method.
           - It is unclear how the hyperparameter is chosen for the proposed method. How does the \beta affect the visualization quality? It would be great to show a range of samples from high to low beta values. Does it require tuning for different visualization samples? Does it vary over different datasets?

---

> ### Author Response · Authors · 2017-12-30
> **Scaling to ImageNet and the Role of the Beta Hyperparameter**
>
> Before directly addressing their concerns, we direct the reviewer’s attention to the newly added Sections E, F, & G in the supplementary material of our revised paper draft. These sections include new experiments that illustrate the effect of varying the beta hyperparameter, demonstrate the strength of our approach on the larger scale Inception network for the ILSVRC 2014 classification challenge, and further highlight the effectiveness of our approach in diagnosing model failure modes.
>
> The phrase “simple datasets” is difficult to interpret; all datasets used in this paper are standard benchmark datasets in computer vision and NLP. We share the reviewer’s desire to further analyze the strength of our framework within computer vision however, for this initial outline of our framework, we have opted to showcase breadth across modalities instead of depth. That said, please consult Section F of our supplementary materials to see visualizations of attention masks trained on top of the Inception network architecture for ImageNet classification. Notably, the results demonstrate that our sample-specific attention masks identify regions of the input space critical to correct classification.
>
> Repeating from a separate comment, LANs trained with an insufficiently large value of beta would accurately reproduce F network outputs without providing useful attention masks; conversely, overly large values of beta dismiss too much information in the input and make reproducing the original network outputs incredibly difficult. Please refer to Section E of the supplementary material for some visualizations illustrating the effect of beta on the resulting attention masks. In general, we make a default assumption that there is a single, if not small range of, beta value that can adequately produce the latent attention mechanisms of the pre-trained network. The parameter does require tuning for different models and, accordingly, we utilize different values of beta across our experiments.

---

### Official Review · AnonReviewer2 · 2017-11-29
**Attention masks for diagnosing neural nets**

**Rating:** 7
**Confidence:** 4

**Review:**

The paper presents the formulation of Latent Attention Masks, which is a framework for understanding the importance of input structure in neural networks. The framework takes a pre-trained network F as target of the analysis, and trains another network A that generates masks for inputs. The goal of these masks is to remove parts of the input without changing the response of F. Generated masks are helpful to interpret the preferred patterns of neural networks as well as diagnose modes of error.

The paper is very well motivated and the formulation and experiments are well presented too. The experiments are conducted in small benchmarks and using simple fully connected networks. It would be interesting to report and discuss convergence properties of the proposed framework. Also, insights of what are the foreseeable challenges on scaling up the framework to real world scenarios.

---

> ### Author Response · Authors · 2017-12-30
> **Role of the Beta Hyperparameter**
>
> Before directly addressing their concerns, we direct the reviewer’s attention to the newly added Sections E, F, & G in the supplementary material of our revised paper draft. These sections include new experiments that illustrate the effect of varying the beta hyperparameter, demonstrate the strength of our approach on the larger scale Inception network for the ILSVRC 2014 classification challenge, and further highlight the effectiveness of our approach in diagnosing model failure modes.
>
> In general, we found that the setting of the beta hyperparameter, weighting the amount of input corruption against the reconstructing the outputs of the pre-trained F network, was the single critical factor in determining convergence to good mask structures. LANs trained with an insufficiently large value of beta would accurately reproduce F network outputs without providing useful attention masks; conversely, overly large values of beta dismiss too much information in the input and make reproducing the original network outputs incredibly difficult. Fortunately, since our approach does not require the re-training of the original network altogether, the grid search over potential beta values is relatively simple, though results must be evaluated qualitatively. Please refer to Section E of the supplementary material for some visualizations illustrating the effect of beta on the resulting attention masks. Accordingly, a fruitful direction for subsequent research involves identifying metrics or alternate loss functions that can better measure the interpretability of the resulting masks thereby minimizing the amount of manual inspection needed to diagnose pre-trained models within our framework.

---

### Decision · Program_Chairs · 2018-01-29
**ICLR 2018 Conference Acceptance Decision**

**Decision:**

Reject

**Comment:**

The proposed LAN provides a visualization of the selectivity of networks to its inputs. It takes a trained network as golden target and estimates an LAN to predict masks that can be applied on inputs to generate the same outputs.
But the significance of the proposed method is unclear, "what is the potential usage of the model?". Empirical justification of that would make it stronger.